# A Proteomic Study of the Bioactivity of *Annona muricata* Leaf Extracts in HT-1080 Fibrosarcoma Cells

**DOI:** 10.3390/ijms241512021

**Published:** 2023-07-27

**Authors:** Ana Dácil Marrero, Ana R. Quesada, Beatriz Martínez-Poveda, Miguel Ángel Medina, Casimiro Cárdenas

**Affiliations:** 1Departamento de Biología Molecular y Bioquímica, Facultad de Ciencias, Universidad de Málaga, Andalucía Tech, E-29071 Málaga, Spain; anadacil@uma.es (A.D.M.); quesada@uma.es (A.R.Q.); bmpoveda@uma.es (B.M.-P.); ccg@uma.es (C.C.); 2Instituto de Investigación Biomédica y Plataforma en Nanomedicina-IBIMA Plataforma BIONAND (Biomedical Research Institute of Málaga), E-29071 Málaga, Spain; 3CIBER de Enfermedades Raras (CIBERER), Instituto de Salud Carlos III, E-28029 Madrid, Spain; 4CIBER de Enfermedades Cardiovasculares (CIBERCV), Instituto de Salud Carlos III, E-28029 Madrid, Spain; 5Research Support Central Services (SCAI), University of Málaga, E-29071 Málaga, Spain

**Keywords:** ultra-high-performance liquid chromatography–high-resolution mass spectrometry (UHPLC–HRMS), *Annona muricata*, graviola, soursop, cancer, cell cycle, iron metabolism, ferroptosis

## Abstract

Graviola (*Annona muricata*) is a tropical plant with many traditional ethnobotanic uses and pharmacologic applications. A metabolomic study of both aqueous and DMSO extracts from *Annona muricata* leaves recently allowed us to identify dozens of bioactive compounds. In the present study, we use a proteomic approach to detect altered patterns in proteins on both conditioned media and extracts of HT-1080 fibrosarcoma cells under treatment conditions, revealing new potential bioactivities of *Annona muricata* extracts. Our results reveal the complete sets of deregulated proteins after treatment with aqueous and DMSO extracts from *Annona muricata* leaves. Functional enrichment analysis of proteomic data suggests deregulation of cell cycle and iron metabolism, which are experimentally validated in vitro. Additional experimental data reveal that DMSO extracts protect HT-1080 fibrosarcoma cells and HMEC-1 endothelial cells from ferroptosis. Data from our proteomic study are available via ProteomeXchange with identifier PXD042354.

## 1. Introduction

Graviola (*Annona muricata*) is an American tropical plant that produces an edible fruit also called graviola, guanábana, guyabano or soursop. Graviola is a source of many bioactive compounds related to its traditional ethnopharmacological uses [1]. Various extracts derived from different parts of the graviola plant, such as the roots, leaves, or fruit, have been extensively studied for their diverse biological activities, which have been comprehensively reviewed elsewhere [1,2,3,4,5]. These activities include antibiotic, antioxidant, anti-inflammatory and antitumoral effects. Notably, the acetogenins present in graviola have been investigated as potential compounds against SARS-CoV-2 [6]. Recently, our research group has demonstrated the anti-angiogenic activity of graviola leaf extracts [7].

In contrast to the extensive information available regarding the ethnopharmacological uses and bioactivities of graviola, limited studies have been conducted using omics approaches. Interestingly, a functional proteomics study showed that graviola leaf ethanolic extract induced endoplasmic reticulum stress and apoptosis in HepG2 hepatocarcinoma cells [8]. Additionally, in our recent contribution showing the anti-angiogenic effects of aqueous and DMSO extracts of graviola leaves, we also carried out a complete, non-targeted metabolomic characterization of these leaf extracts [7].

Provided the reported effects of graviola on cancer models and tumoral cells, and aiming to reveal new possible effects, we carried out a proteomic study on cultured HT-1080 fibrosarcoma cells. In addition, taking advantage of our expertise in the field of angiogenesis, and based on our previous results on extracts of this plant as modulators of this process, in which endothelial cells play a fundamental role, we also used the human endothelial cell line HMEC-1 as a non-tumoral line in certain experiments.

## 2. Results

### 2.1. HT-1080 Fibrosarcoma Cells and HMEC-1 Survival Curves after Treatment with Annona muricata Extracts

We have previously shown that *A. muricata* extracts exhibit low toxicity against bovine aorta endothelial cells (BAEC), with IC_50_ values of 983 ± 345 ng/mL and 962 ± 38 µg/mL for the DMSO and the aqueous extracts, respectively [7]. Figure 1 shows new data obtained with the MTT assay in HT-1080 human fibrosarcoma cells and HMEC-1 human microvascular endothelial cells. As in the case of BAEC, both extracts exhibited low cytotoxicity against HT-1080 fibrosarcoma cells, with the IC_50_ value for DMSO extracts being two orders of magnitude lower than that for aqueous extracts. Similar results were obtained with HMEC-1.

### 2.2. Deregulated Proteins in HT-1080 Fibrosarcoma Cells after Treatment with Annona muricata Extracts

In a previously published study, we showed that *A. muricata* extracts exhibit anti-angiogenic properties [7]. This was a novel biological property to be added to a long list of previously described biological effects of *A. muricata* extracts. In the present study, to obtain a deeper insight into new biological effects of *A. muricata* extracts, we carried out a detailed proteomic analysis. Samples for the proteomic analysis were obtained as described in Materials and Methods (Section 4.5 and Section 4.6) from HT-1080 fibrosarcoma cells treated with 0.1 mg/mL of aqueous extracts or 0.01 mg/mL of DMSO extracts for 24 h. These doses were one order of magnitude lower than the respective IC_50_ values, thus precluding the possible interference of cytotoxic effects. Proteomic analysis was carried out as described in Materials and Methods (Section 4.7, Section 4.8, Section 4.9, Section 4.10).

The lists of all significantly over- and under-expressed proteins (*p*-value < 0.05) in HT-1080 cell extracts and conditioned media after treatment with aqueous or DMSO extract from *A. muricata* are shown in Appendix A. Treatments with aqueous extracts produced the overexpression of 32 proteins and the underexpression of 49 proteins in the analyzed HT-1080 cell extracts (Appendix A). The figures were 9 and 6, respectively, in the case of HT-1080 conditioned media (Appendix A). Treatments with DMSO extracts produced the overexpression of 15 proteins and the underexpression of 30 proteins in the analyzed HT-1080 cell extracts (Appendix A). The figures were 3 and 5, respectively, in the case of HT-1080 conditioned media (Appendix A). Deregulated proteins are represented in volcano plots in Figure 2A, where over and underexpressed proteins with *p*-value < 0.01 are shown within red and green rectangles, respectively.

Figure 2B depicts scatter plots of deregulated proteins in both HT-1080 fibrosarcoma cell extracts and conditioned media after treatment with *A. muricata* aqueous or DMSO extracts. This representation of the data, where results are referred to their control condition, facilitates the comparison between treatments. Among the proteins with a significant change in their levels, two components of ferritin, ferritin light chain (FTL) and ferritin heavy chain (FTH1), are reduced with both treatments, the first in cell extracts, and the second in the conditioned media (Figure 2B). In the same direction, a cell surface hyaluronidase (CEMIP2) is also reduced in cell extracts after both treatments. In addition, WD repeat-containing protein 26 (WDR26) is overexpressed in cell extracts of HT-1080 treated with aqueous and DMSO extracts (Figure 2B). Eventually, cyclin-dependent kinase 4 (CDK4) is reduced in cell extracts after treatment with DMSO extract (Appendix A). These results will be further commented on and discussed.

Figure 3 shows some data obtained from the STRING functional enrichment analysis. With a high level of confidence (interaction score ≥ 0.7), only three small subnetworks with 2, 3 and 4 nodes were highlighted in the case of downregulated proteins from samples coming from cell extracts of HT-1080 treated with DMSO extracts of *A. muricata*. None of the networks revealed by STRING analysis from the other experimental conditions significantly exhibited more interactions than expected. Among the 29 downregulated proteins in cell extracts of HT-1080 treated with DMSO extracts of *A. muricata*, Figure 3 shows that there were seven interactions when the expected number of interactions considering the whole proteome was three. The proteins that participate in the interaction networks are non-SMC condensin I complex subunit D2 (NCAPD2), structural maintenance of chromosomes 2 (SMC2) and 4 (SMC4), RUNX family transcription factor 1 (RUNX1), thrombospondin 1 (THBS1), secreted protein acidic and cysteine rich (SPARC), histone acetyltransferase 1 (HAT1) and mevalonate kinase (MVK). Remarkably, interactions between NCAPD2, SMC2, SMC4 and HAT1 are involved in chromosome condensation, suggesting potential effects on the HT-1080 cell cycle. Another process identified in the analysis is lipid biosynthesis, in which MVK and CYP51A1 are involved.

### 2.3. Annona muricata DMSO Extract Affects Cell Cycle Distribution of HT-1080 Cell Subpopulations

The proteomic study revealed that CDK4 was significantly reduced in HT-1080 cells under treatment with DMSO extracts of *A. muricata* leaves (Figure 4A, bottom panel, and Appendix A). Further STRING analysis also suggested a cell cycle-related functional enrichment (Figure 3). Provided this evidence, we turned to the proteomic study to check the levels of another cyclin-dependent kinase, cyclin-dependent kinase-1 (CDK1) and, although not included among the proteins with significant changes (*p* < 0.05), we did detect a significant decrease in CDK1 levels in cell extracts compared to the control (Figure 4A, upper panel). Then, an independent analysis of the distribution of cell subpopulations in the different phases of the cell cycle was performed, clearly showing that the reduction in the amount of CDK4 induced by treatment with DMSO extracts of *A. muricata* leaves is accompanied by a relative increase in the G_0_/G_1_ subpopulation, suggesting an arrest in this cell cycle phase (Figure 4B,C).

### 2.4. Annona muricata Aqueous Extract Modulates the Levels of Several Components of Iron Metabolism in HT-1080 Cells

The proteomic analysis showed decreased levels of ferritin (FTL, ferritin light chain; FTH1, ferritin heavy chain) in both cell extracts and conditioned media. Additionally, functional enrichment analysis evidenced that treatment with aqueous extracts of *A. muricata* leaves affected the levels of some components of iron metabolism, as revealed by the KEGG ferroptosis pathway (Figure 5A). Transferrin receptor 1 (TFR1) levels were increased whereas transferrin and ferritin levels were decreased, as compared with control, non-treated cells. Two of these data were validated by independent analysis; namely, increased levels of TFR1 were determined by Western blot analysis (Figure 5B,C), and decreased ferritin was validated by immunoassay (Figure 5D).

### 2.5. Annona muricata DMSO Extract Protects HT-1080 Cells from Ferroptosis

The changes in the levels of proteins involved in iron metabolism mentioned in Section 2.4 suggest that aqueous extracts of *A. muricata* leaves could affect ferroptosis in HT-1080 fibrosarcoma cells. Appendix A shows that neither aqueous nor DMSO extracts of *A. muricata* leaves were able to induce ferroptosis in HT-1080 cells. Figure 6 shows that DMSO (but not aqueous) extracts of *A. muricata* leaves were able to rescue both HT-1080 fibrosarcoma and HMEC-1 human microvascular endothelial cells from ferroptosis induced by erastin, a ferroptosis-inducer molecule [9].

## 3. Discussion

In our previously published metabolomic study of *A. muricata* extracts, we showed that these extracts have anti-angiogenic properties [7]. This was a novel biological property to be added to a long list of previously described biological effects of *A. muricata* extracts, including antileishmanial, mulloscicidal, antibacterial, anti-parasitic, anti-inflammatory, antitumoral, and anti-SARS-CoV2 activities [2,3,4,5,6,10,11,12,13,14,15]. Moreover, proteomic studies are a great tool to obtain further insight into novel bioactivities of *A. muricata* extracts. However, up to now, few available data have been published on this issue [8]. Our present study makes use of a proteomic approach to identify new effects of these extracts with therapeutic potential. The list of bioactive compounds that we previously identified in these extracts by using HPLC (both with diode array detector and charged aerosol detector) and UHPLC-HMRS [7] is included in Appendix A.

To select the extract doses for the treatments to be submitted to proteomic analysis, we first carried out a cell survival assay with MTT in both fibrosarcoma cells (HT-1080) and an immortalized human microvascular endothelial cell line (HMEC-1). Results of this assay confirmed that both extracts have low cytotoxicity, even milder in the case of aqueous extracts, as previously shown by us for BAEC [7]. For the proteomic study and the validation experiments, we always used doses of extracts without cytotoxic effects.

Our proteomic study and the subsequent functional enrichment analysis (Figure 2, Figure 3 and Figure 5A, and Appendix A) have thrown light on the potential effects of *A. muricata* extracts on cell cycle and iron metabolism. STRING functional enrichment analysis among the significantly downregulated proteins in HT-1080 cells treated with DMSO extract of *A. muricata* also revealed that lipid biosynthesis is affected in HT-1080-treated cells. However, this observation has not been the subject of additional validation in the present study, but it does deserve to be studied.

Regarding the changes in the levels of cell cycle kinases CDK1 and 4, their decreased relative abundance in samples from HT-1080 cell extracts treated with DMSO extracts (Figure 4A) suggested effects on cell cycle subpopulation distribution, which could be validated by flow cytometric analysis making use of propidium iodide (Figure 4B,C). The G_0_/G_1_ subpopulation was significantly increased by treatment, indicating that the DMSO extract of *A. muricata* induced an accumulation of cells halted before the S phase. Therefore, these results suggest that the DMSO extract of *A. muricata* could have cytostatic effects on HT-1080 fibrosarcoma cells. These results are also in line with the detected 20-times fold-change overexpression of WDR26 in cell extracts (Appendix A), as this protein has been described as a negative regulator of the MAPK pathway, which directly regulates survival and proliferation in some cell types [16]. Interestingly, other studies have proved that different *A. muricata* extracts and fractions exert similar cytostatic effects in other tumoral cells, such as breast and colon cancer cells, specifically on the G_0_/G_1_ phase [17,18]. This evidence also agrees with the cytostatic and anticancer effects of graviola plant extracts that have been extensively described.

Regarding the effects of aqueous extracts of *A. muricata* on HT-1080 fibrosarcoma cell iron metabolism, validation studies revealed an increase in transferrin receptor TFR1 and a decrease in the intracellular levels of ferritin (Figure 5). These data could suggest effects on ferroptosis. Ferroptosis is a metabolically regulated cell death process. In cancer cells, metabolism undergoes significant rewiring to meet their increased energy and synthesis requirements and support their rapid proliferation. This metabolic reprogramming often leads to unique metabolic characteristics, including an abundance of phospholipids with high content of poly-unsaturated fatty acids, especially sensitive to oxidation, excessive iron accumulation and imbalanced defense systems against ferroptosis [19]. These features present a promising opportunity to identify new therapeutic targets in cancer that exploit the vulnerability of cancer cells to ferroptosis, sensitizing them. Thus, combining ferroptosis-inducing agents with conventional therapies that promote ferroptosis can enhance therapeutic effectiveness, as evidenced by synergistic effects and tolerability in preclinical models [20,21]. In fact, this approach is already being evaluated in clinical studies [22,23,24].

Nonetheless, a comprehensive evaluation of histological and pharmacological parameters remains crucial to assess the potential toxic effects of ferroptosis inducers on normal tissues and determine optimal drug dosage. In fact, besides cancer cells, various cell types within the tumor microenvironment, including immune cells that can either enhance or inhibit antitumor immune responses, may also exhibit susceptibility to ferroptosis. Notably, immune cells are especially sensitive to ferroptosis [19]. This explains why not all types of cancer will potentially benefit from combined therapy with ferroptosis-inducing agents. In inflamed tumors, or “hot tumors”, where high infiltration of different immune cells occurs, this combinatory therapy would be detrimental, as these immune cells will potentially die, which could promote tumorigenesis. Conversely, immune-excluded tumors, or “cold tumors”, with low infiltration of immune cells, are the ones who benefit from treatment with ferroptosis-inducing agents [25,26].

Our results show that *A. muricata* aqueous extracts have no impact on ferroptosis (Appendix A), and it does not rescue from erastin-induced ferroptosis (Figure 6). In contrast, *A. muricata* DMSO extracts rescued HT-1080 fibrosarcoma cells from erastin-induced ferroptosis (Figure 6A,B).

The mechanism by which ferroptosis protectors or “rescuers” play their role usually involves cellular antioxidant systems that directly neutralize lipid peroxides, such as the GPX4–GSH, FSP1–CoQH2, DHODH–CoQH2 or GCH1–BH4 systems [19]. These systems might be participating in the mechanism of action of *A. muricata* DMSO extract as well, but more studies are needed to determine this. Notably, the protein WDR26 has also shown potent antioxidant properties in certain cell types [27], which could also explain the anti-ferroptosis effect of this plant extract. Remarkably, the use of these ferroptosis inhibitors is an attention-grabbing therapeutical approach that has not been well explored. In the case of inflamed tumors, as traditional chemotherapy can also induce ferroptosis in cells of the immune system, the use of ferroptosis “rescuers” could boost anticancer immunity by preventing immune cell death. However, fine regulation of drug type, dose and specificity, must be determined to specifically protect immune cells from ferroptosis, and not cancer cells.

Furthermore, we have proved that *A. muricata* DMSO extract “rescues” endothelial cells from ferroptosis (Figure 6C,D). Remarkably, anti-ferroptosis therapy is a rather promising approach to restoring endothelial dysfunction, strongly correlated with the initiation and progression of many vascular conditions such as atherosclerosis, blood–brain barrier injury or ischemia/reperfusion injury [28,29,30]. In such pathological situations, it is usual that cell death mechanisms such as pyroptosis or ferroptosis are strongly active due to prolonged exposure to damaging stimuli, e.g., ROS [31,32]. Interestingly, recent work has put the focus on ferroptosis in cardiomyocytes as a specific mechanism of heart failure, which further supports the importance of these ferroptosis “rescuers” [33].

Regarding the composition of graviola extracts, which has been extensively studied, and was also described in our previous article [7], it is challenging to determine whether the bioactive effects are primarily attributable to the action of a few compounds, a group of them or the synergistic action of a vast majority. In fact, only a few isolated compounds from *A. muricata* have been investigated for their biological and pharmacological activities, especially anti-inflammatory and anticancer activities [1,2,3,4,5]. Interestingly, it has been described that Annonaceous acetogenins (AGEs), alkaloids and phenols are the bioactive metabolites isolated from the *A. muricata* leaves, and among them, AGEs are the most predominant [4]. Focusing on the results of this study, it is worth mentioning that molecules such as kaempferol, quercetin and chlorogenic acid, all present in *A. muricata* extracts, protected different cellular specimens from ferroptosis [34,35,36,37], suggesting that this effect might be due to the cumulative individual activities of the different compounds and their synergistic action.

To summarize, two interesting bioactivities of *A. muricata* DMSO extract were revealed in this study, namely cytostatic effects on tumor cells and ferroptosis “rescue” of tumor and endothelial cells. Induction of ferroptosis stands as a great opportunity to promote cancer cell death in a combinatorial therapy with traditional chemotherapy. However, only tumors with low infiltration of immune cells, the so-called “cold tumors”, are applicable to this approach, as immune cells are highly sensitive to the induction of ferroptosis, which could be detrimental to antitumoral immunity. In those cases, ferroptosis “rescuers” are an attractive strategy to protect immune cells from undergoing ferroptosis, that has been little explored. In other contexts, these ferroptosis inhibitors are currently being investigated as a compelling therapeutic approach in pathologies associated with endothelial dysfunction, such as atherosclerosis, wherein endothelial cells exhibit intensified susceptibility to ferroptosis due to prolonged exposure to oxidative stimuli. In this work, we have demonstrated that *A. muricata* extract in DMSO protects cells from ferroptosis similarly to ferrostatin-1, which brands this natural extract an intriguing candidate for boosting anticancer immunity and enhancing endothelial function. However, further preclinical studies on cell cultures and animal models are needed to unravel the mechanism of action of this substance derived from the graviola plant, as well as to identify the specific compounds within this extract that are responsible for its intriguing pharmacological effects. Furthermore, it is also essential to determine whether the combined action of these compounds yields superior results compared to their individual effects. Such investigations could establish the groundwork for future clinical studies, potentially leading to a novel anti-ferroptosis therapy with applications for specific tumors or in diseases that course with endothelial dysfunction.

## 4. Materials and Methods

### 4.1. Plant Material

Dried graviola leaf powder was purchased from Tentorium Energy SL (Tarragona, Spain). To prepare the aqueous extract, 25 g of graviola powder was weighed and added to 500 mL of MilliQ water, incubated in a water bath at 80 °C for 10 min, centrifuged at 13,000× *g*. The supernatant was recovered, filtered and frozen at −80 °C, lyophilized and reconstituted with sterile water to a final concentration of 1 mg/mL. The DMSO extract was prepared by weighing 1 g of graviola powder and resuspending it in 10 mL of DMSO, then it was incubated for 5 min at room temperature with gentle shaking, centrifuged at 13,000× *g* and the supernatant was recovered and filtered with PTFE filters (Gelman Sciences, Washtenaw, MI, USA). Then, 100 µL of the filtered supernatant was taken, dried under a nitrogen jet and redissolved in acetonitrile to a final concentration of 1 mg/mL. The chromatographic and spectroscopic analyses and characterization of these extracts were previously communicated by us elsewhere [7]. Bioactive compounds identified in these extracts are listed in Appendix A.

### 4.2. Cell Culture

Human HT-1080 fibrosarcoma cells were maintained in Dulbecco’s modified Eagle’s medium (DMEM) containing 4.5 g/L glucose, 2 mM glutamine and penicillin/streptomycin (Corning, Somerville, MA, USA), supplemented with 10% fetal bovine serum (FBS; Capricorn Scientific GmbH, Ebsdorfergrund, Germany). The transformed microvascular endothelial cell line HMEC-1 was kindly supplied by Dr. Arjan W. Griffioen (Maastricht University, The Netherlands). HMEC-1 was cultivated in MCDB-131 (Corning, Somerville, MA, USA) medium supplemented with 1 µg/mL hydrocortisone, 10 ng/mL of EGF-1 (Sigma/Merck, Darmstadt, Germany), 1% penicillin/streptomycin solution, 2 mM L-glutamine and 10% FBS. Cells were kept in a humid incubator at 37 °C under conditions of 5% carbon dioxide. Subconfluent (at 75–80% of confluency) cells were used for subculturing and for treatments and experiments.

### 4.3. MTT Assay

The 3-(4,5-dimethylthiazol-2-yl)-2,5-diphenyltetrazolium bromide (MTT; Sigma/Merck, Darmstadt, Germany) dye reduction assay in 96-well microplates was used. HT-1080 fibrosarcoma cells and HMEC-1 were incubated for 3 days in each well with serial dilutions of aqueous or DMSO *A. muricata* extracts (37 °C, 5% CO_2_ in a humid atmosphere); 10 µL of MTT (5 mg/mL in PBS) was added to each well, and the plate was incubated for a further 4 h (37 °C). The resulting formazan was dissolved in 150 µL of 0.04 N HCl-2-propanol and read at 550 nm. Four samples for every tested concentration were included in each of three independent experiments. IC_50_ values were calculated as those concentrations of *A. muricata* extract yielding a 50% of cell survival, taking the values obtained for control (cells treated with DMSO) as 100%.

### 4.4. Treatments for Proteomic Analysis

For proteomic analysis, HT-1080 cells were treated for 24 h in FBS-free DMEM under three different experimental conditions: untreated controls, treatment with graviola aqueous extract at 0.1 mg/mL and treatment with graviola DMSO extract at 0.01 mg/mL.

### 4.5. Sample Preparation, Protein Extraction and Clean-Up

After treatments, culture media were collected, centrifuged to remove cell debris and supernatants immediately frozen at −80 °C for further lyophilization. Meanwhile, cells were washed with ice-cold phosphate-buffered saline (PBS) and kept frozen at −80 °C in the culture flasks until further processing. Keeping the sample on ice, each cell layer was solubilized in 200 μL of 7 M urea, 2 M thiourea, 4% CHAPS buffer and then sonicated on a cold ultrasound bath for 5 min. Cell extracts were then centrifuged at 14,000× *g* at 4 °C for 5 min to remove insoluble debris.

Proteins from both culture media and cell extracts were purified by a modified trichloroacetic acid protein precipitation procedure (Clean-Up Kit; GE Healthcare, München, Germany). The resulting protein pellets were dissolved in 100 μL of water. After thoroughly vortexing the samples, they were sonicated on an ultrasound bath for 5 min and centrifuged at 14,000× *g* for 5 min; the supernatant was transferred to a clean tube. Finally, sample concentration was quantified by a Bradford assay [38].

### 4.6. In-Gel Digestion and Peptide Extraction

We carried out a gel-assisted proteolysis, entrapping the protein solution in a polyacrylamide gel matrix. Samples (45 μL) were thoroughly mixed with 14 μL of 40% acrylamide monomer solution, and 2.5 μL of 10% ammonium persulfate and 1 μL of TEMED were quickly added. The mixture was allowed to completely polymerize for 20 min before performing an in-gel digestion. Using scalpel, gel plugs were cut into 1–2 mm cubes and treated with 50% acetonitrile/25 mM ammonium bicarbonate. Samples were dehydrated and desiccated with acetonitrile (ACN) before reduction with 10 mM dithiothreitol (DTT) in 50 mM ammonium bicarbonate for 30 min at 56 °C. Excess DTT was removed, and cysteine residues were carbamidomethylated with 55 mM iodoacetamide in 50 mM ammonium bicarbonate for 20 min at room temperature in the dark. After carbamidomethylation, the gel pieces were dehydrated again; proteins were digested by rehydrating the gel pieces in trypsin solution at 10 ng/μL (Pierce trypsin protease, MS grade; Thermo Fisher Scientific, Waltham, MA, USA) and thereafter incubated at 30 °C overnight. Peptides were extracted from the gel pieces with ACN/0.1% formic acid (FA) for 30 min at room temperature. The samples were dried in a SpeedVac^TM^ (Thermo Fisher Scientific, Waltham, MA, USA) vacuum concentrator to remove ACN and residual ammonium bicarbonate, redissolved in 50 μL of 0.1% FA, sonicated for 3 min and centrifuged at 14,000× *g* for 5 min. Finally, the samples were quantified again in a NanoDrop^TM^ (Thermo Fisher Scientific, Waltham, MA, USA); 0.1% FA was added to equalize all samples at an identical protein concentration before being transferred to the injection vial.

### 4.7. Liquid Chromatography High-Resolution Mass Spectrometry (HPLC-MS)

Samples were injected onto an Easy nLC 1200 UHPLC system coupled to a hybrid linear trap quadrupole Orbitrap Q-Exactive HF-X mass spectrometer (Thermo Fisher Scientific, Waltham, MA, USA). Software versions used for the data acquisition and operation were Tune 2.9 and Xcalibur 4.1.31.9. HPLC solvents were as follows: solvent A consisted of 0.1% formic acid in water, and solvent B consisted of 0.1% formic acid in 80% acetonitrile. From a thermostated autosampler, 2 μL that correspond to 100 ng of the peptide mixture were automatically loaded onto a trap column (Acclaim PepMap 100, 75 μm × 2 cm, C18, 3 μm, 100 A, Thermo Fisher Scientific, Waltham, MA, USA) at a flow rate of 20 μL/min and eluted onto a 50 cm analytical (PepMap RSLC C18, 2 μm, 100 A, 75 μm × 50 cm, Thermo Fisher Scientific, Waltham, MA, USA). The peptides were eluted from the analytical column with a 120 min gradient ranging from 2% to 20% solvent B, followed by a 30 min gradient from 20% to 35% solvent B and finally, to 95% solvent B for 15 min before re-equilibration to 2% solvent B at a constant flow rate of 300 nL/min. The LTQ Velos ESI Positive Ion Calibration Solution (Thermo Fisher Scientific, Waltham, MA, USA) was used to externally calibrate the instrument prior to sample analysis, and an internal calibration was performed on the polysiloxane ion signal at *m*/*z* 445.120024 from ambient air. MS1 scans were performed from *m*/*z* 300–1750 at a resolution of 120,000. Using a data-dependent acquisition mode, the 20 most intense precursor ions of all precursor ions with +2 to +5 charge were isolated within a 1.2 *m*/*z* window and fragmented to obtain the corresponding MS/MS spectra. The fragment ions were generated in a higher energy collisional dissociation (HCD) cell with a fixed first mass at 110 *m*/*z* and detected in an Orbitrap mass analyzer at a resolution of 30,000. The dynamic exclusion for the selected ions was 30 s. Maximal ion accumulation time allowed in MS and MS^2^ mode was 50 ms. Automatic gain control was used to prevent overfilling of the ion trap and was set to 3 × 10^6^ ions and 10^5^ ions for a full MS and MS^2^ scan, respectively.

### 4.8. Data Analysis for Protein Identification

MS/MS spectra were searched against SwissProt *Homo sapiens* protein database canonical version 2021.09.30 (20,315 sequences). The UniProt *Annoma muricata* database (117 sequences, version 2023-05-03) and the UniProt *Arabidopsis Thaliana* database (119,635 sequences, version 2021-09-01) were used as plant contaminant databases. The acquired raw data were analyzed in Proteome Discoverer^TM^ 2.5 (Thermo Fisher Scientific, Waltham, MA, USA) platform with the SEQUEST^®^ HT engine using mass tolerances of 10 ppm and 0.02 Da for precursor and fragment ions, respectively. Two missed tryptic cleavage sites were allowed. Oxidation of methionine and N-terminal acetylation were set as variable modifications, whilst carbamidomethylation of cysteine residues, was set as fixed modification. Peptide spectral matches (PSM) and consecutive protein assignments were validated using the Percolator^®^ algorithm [39], based on a target-decoy approach using a reversed protein database as the decoy by imposing a strict cut-off of 1% false discovery rate (FDR). Peptide identifications were grouped into proteins according to the law of parsimony, and results were filtered to contain only proteins with at least two unique peptide sequences.

### 4.9. Label-Free Relative Quantification for Differential Expression Analysis

Label-free quantitation was implemented using the Minora feature of Proteome Discoverer^TM^ 2.5 (Thermo Fisher Scientific, Waltham, MA, USA), setting the following parameters: maximum retention time alignment of 10 min with minimum S/N of 5 for feature linking mapping. Abundances were based on precursor intensities. Normalization was performed based on total peptide amount, and samples were scaled on all averages (for every protein and peptide, the average of all samples is 100). The normalized and scaled relative abundance of every protein was expressed as mean ± standard deviation (SD) of three biological replicates. Protein abundance ratios were directly calculated from the grouped protein abundances. Abundance ratio *p*-values were calculated by ANOVA based on the abundances of individual proteins or peptides. Only proteins with ANOVA *p* < 0.01 and higher ratio than 2:1 or smaller than 1:2 for treatment:control were considered as significantly deregulated. The MS proteomics data have been deposited to the ProteomeXchange Consortium [40] via the PRIDE partner repository with the dataset identifier PXD042354.

### 4.10. Bioinformatic Analysis

The Search Tool for the Retrieval of Interacting Genes/Proteins (STRING) (https://www.string-db.org, accessed on 21 April 2023) was used to perform a protein–protein interaction networks functional enrichment analysis. Functional pathways were analyzed using the Kyoto Encyclopedia of Genes and Genomes (KEGG) database (http://www.genome.jp/kegg/pathway.html, accessed on 29 November 2022).

### 4.11. Cell Cycle Subpopulation Distribution

HT-1080 cells in 6-well plates at 70–80% confluence were incubated overnight in the presence or absence of *A. muricata* DMSO extract (10 or 100 µg/mL). A negative (DMSO) and positive control of cell impairment (10 μM 2-methoxyestradiol) were included. After overnight incubation, cells were collected and washed with PBS and then permeabilized with ice-cold 70% ethanol for 1 h. Permeabilized cells were then incubated with 100 μg/mL RNAse (Sigma/Merck, Darmstadt, Germany) and 40 μg/mL propidium iodide (Sigma/Merck, Darmstadt, Germany) for 30 min at 37 °C protected from light. The percentages of cells in the G_0_/G_1_, S, and G_2_/M phases of the cycle, and the population in sub-G_1_ (fragmented DNA), were determined using a BD Biosciences FACS VERSETM flow cytometer (Becton Dickinson, Franklin Lakes, NJ, USA). The resulting data were analyzed with the Kaluza software (Beckman Coulter, Brea, CA, USA).

### 4.12. Western Blot Analysis

HT-1080 cells in 6-well plates at 70–80% confluence were incubated overnight in the presence or absence of *A. muricata* aqueous extract 100 µg/mL. Then, conditioned media was collected, and cells were lysed in 100 µL of RIPA lysis buffer (Sigma/Merck, Darmstadt, Germany). The protein concentration of the samples was estimated using a Bradford assay [38], and a volume corresponding to 30 μg of total protein of cell lysates, and the equivalent volume of conditioned media, were subjected to SDS-PAGE denaturing electrophoresis.

After electrophoresis, gels were electrotransferred to a nitrocellulose membrane. Membranes were blocked in TBS-T buffer (20 mM Tris, 137 mM NaCl, 0.1% Tween-20) containing 5% semi-skimmed milk and then incubated overnight with rabbit monoclonal anti-transferrin receptor (Cell Signaling Technology, Danvers, MA, USA) antibody diluted 1:500–1000 in TBS-T with 5% BSA. After incubation with the secondary antibody diluted 1:5000 in blocking buffer, the signal was detected using the SuperSignal West Picochemiluminescence system (Thermo Fisher Scientific, Waltham, MA, USA) and an imaging system Chemidoc XRS (Bio-Rad, Hercules, CA, USA). The same membranes were incubated with the anti-tubulin antibody at a dilution of 1:1000. Blots were quantified by densitometry with the software FIJI.

### 4.13. Ferritin Quantification

Ferritin quantification analysis was carried out by making use of a two-point sandwich immunoassay by direct chemiluminometric technology, which uses constant amounts of two anti-ferritin antibodies, using an Atellica™ IM Analyzer (Siemens Healthineers, Erlangen, Germany) [41].

### 4.14. Ferroptosis Rescue Assay

HT1080 or HMEC-1 cells in 6-well plates at 70–80% confluence were incubated overnight with the ferroptosis-inducer erastin alone or in cotreatment with ferrostatin-1 (used as a positive control of ferroptosis rescue), *A. muricata* DMSO extract 10 or 100 µg/mL, or *A. muricata* aqueous extract 100 µg/mL. A negative control (DMSO) without erastin was also included. 20 µM of erastin and 5 µM of ferrostatin-1 (both from Sigma/Merck, Darmstadt, Germany) were used. After the incubation, cells were incubated with propidium iodide (1 µg/mL) for 15 min at 37 °C, and both bright field and fluorescence pictures of the cells in the different conditions were taken with a Nikon DSRi2 camera attached to a Nikon Eclipse Ti microscope (Nikon, Minato, Japan). Images were analyzed with the FIJI software.

### 4.15. Statistical Analysis

The results are shown as the mean value of at least three independent replicates and their corresponding standard deviation (SD) values. Statistical significance was determined by *t*-test or one-way ANOVA and Dunnett’s multiple comparisons test; values of *p* < 0.05 were considered statistically significant. Significance was indicated as follows: * *p* < 0.05, ** *p* < 0.01, *** *p* < 0.001, **** *p* < 0.0001. Graphpad Prism 9 was used for the statistical analysis.

## Figures and Tables

**Figure 1 ijms-24-12021-f001:**
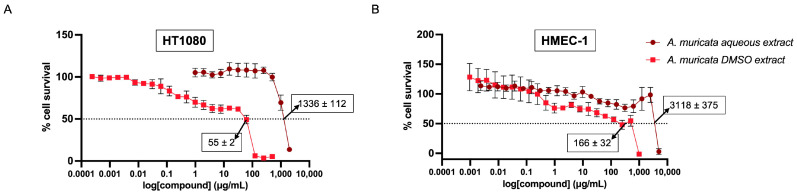
Effect of *Annona muricata* extracts on cell survival. The figure shows the survival curves of HT-1080 (**A**) and HMEC-1 (**B**) cells after 72 h treatment with increasing doses of the *A. muricata* aqueous and DMSO extracts. The point where the dashed line intersects the survival curve indicates the IC_50_ (μg/mL), meaning the concentration that allows 50% of culture survival, pointed by arrows. Data are provided as means ± SD of three independent experiments.

**Figure 2 ijms-24-12021-f002:**
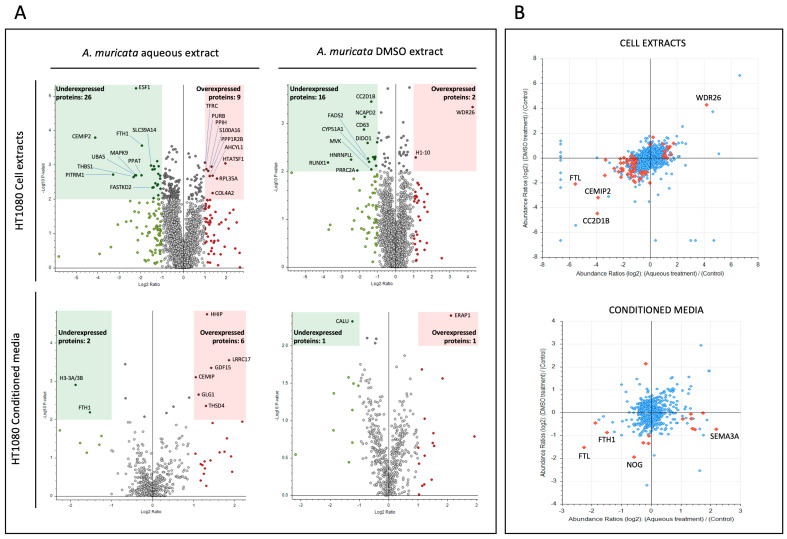
Differential protein expression in HT-1080 treated with *Annona muricata* extracts. (**A**) Volcano plots show deregulated proteins in HT-1080 cell extracts and conditioned media after treatment with *A. muricata* aqueous and DMSO extracts. Proteins were considered upregulated (red square) or downregulated (green square) if fold-change values were >2 (red circles) or <0.5 (green circles), respectively. Among these proteins, changes in those with a *p*-value < 0.01 were considered significant and grouped in red and green squares. The top ten deregulated proteins are labeled in each case. Gray circles, not significant. (**B**) Scatter plots show the Log2 protein abundance ratio when HT-1080 cells were treated with *A. muricata* DMSO extract versus aqueous extract in both cell extracts and conditioned medium. Up and down-regulated proteins (log2 abundance ratio > 1 and < −1, respectively and *p*-value < 0.01 are represented as red rhombuses.

**Figure 3 ijms-24-12021-f003:**
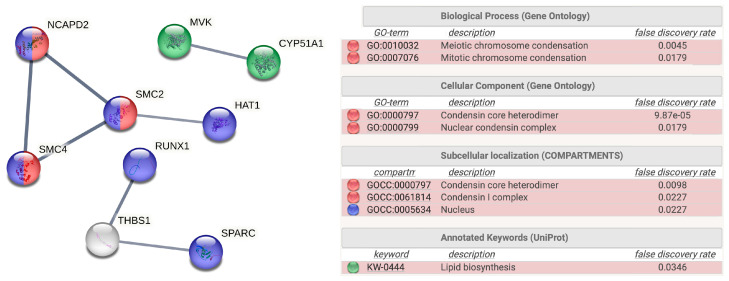
STRING analysis of deregulated proteins. Protein–protein interaction networks functional enrichment analysis among the significantly downregulated proteins in HT-1080 cells treated with DMSO extract of *Annona muricata* (fold change < 0.5, *p*-value < 0.05) using STRING, the Search Tool for the Retrieval of Interacting Genes/Proteins. Network nodes are proteins, and edges represent the predicted functional associations. The thickness of a line indicates the strength of the interaction between the proteins it connects. Colors refer to a representative partial list of the significantly enriched Gene Ontology (GO) terms and annotated keywords in UniProt database. Network stats and functional enrichment details are given in Appendix A.

**Figure 4 ijms-24-12021-f004:**
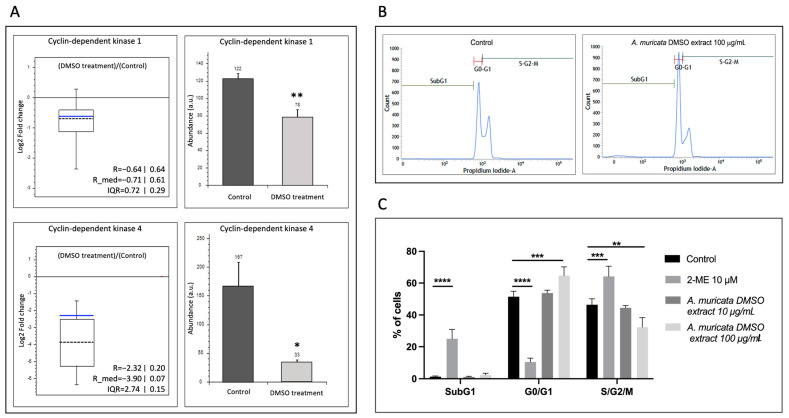
*Annona muricata* DMSO extract affects cell cycle of HT-1080 cells. (**A**) Left: ratio distribution of DMSO extract treatment versus control of all peptides considered for calculating the abundance of CDK1 (up) and CDK4 proteins (down) as a box-and-whisker plot. The peptide group ratios are shown as the binary logarithm. The box represents the peptide group ratios between the 25th and the 75th percentiles. The error bars represent the peptide group ratios below the 5th and the 95th percentiles. The blue line represents the mean of the distribution, and the dashed line the median. The chart displays the mean (R) of the distribution, the median (R_med) and the inter-quartile range (IQR) in linear and logarithmic format. On the right, the same data are presented in a bar chart showing the means ± SD. Statistical differences were assessed by Student’s *t*-test, * *p* < 0.05; ** *p* < 0.01. (**B**) Representative profiles of the cell cycle of HT-1080 in the presence of DMSO (Control condition) or *A. muricata* DMSO extract 100 µg/mL, assayed by flow cytometry. (**C**) Bar diagram showing the percentage of cells in the different phases of the cell cycle in the control condition or treated with 2-methoxyestradiol 10 µM (positive control cell cycle disruption), or *A. muricata* DMSO extract 10 or 100 µg/mL. Means ± SD of a minimum of three independent experiments are shown, and one-way ANOVA + Dunnett’s multiple comparisons test was performed for the statistical analysis (** *p* < 0.01; *** *p* < 0.001; **** *p* < 0.0001).

**Figure 5 ijms-24-12021-f005:**
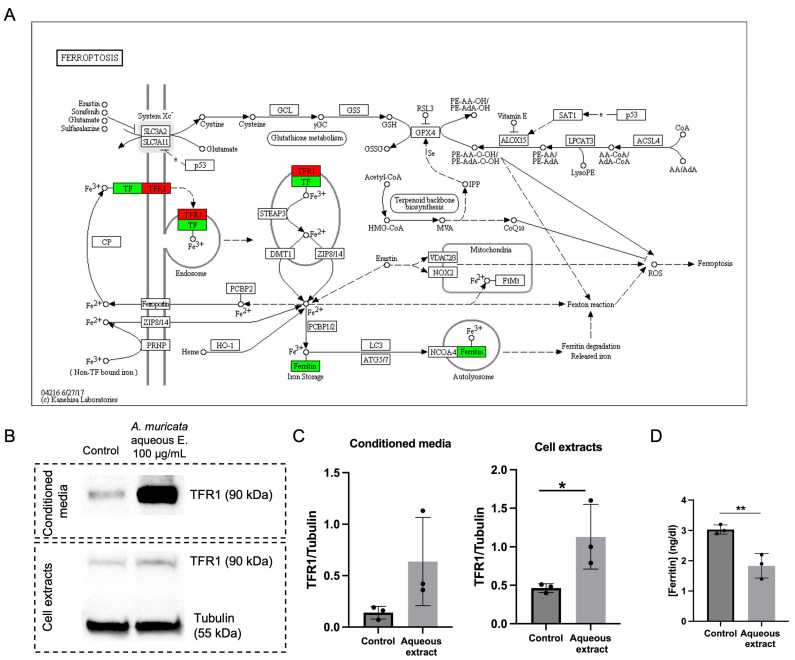
*Annona muricata* aqueous extract modulates the levels of several proteins involved in iron metabolism in HT-1080 cells. (**A**) KEGG ferroptosis pathway map from *Homo sapiens* (pathway: hsa04216) showing overexpressed (red) and underexpressed proteins (green) when HT-1080 cells were treated with aqueous extract of *A. muricata*. The image was obtained by a valid license from the KEGG module within Proteome Discoverer software (Thermo Scientific). (**B**) Western blot densitometry images of the transferrin receptor-1 (TFR1) in the conditioned media and cell extracts of HT-1080 upon aqueous extract treatment. (**C**) Relative quantification of the bands, normalized by tubulin. (**D**) Ferritin detection by a two-point sandwich immunoassay using direct chemiluminometric technology. Mean ± SD of a minimum of three independent experiments are shown, and Student’s *t*-test was performed for the statistical analysis (* *p* < 0.05, ** *p* < 0.01).

**Figure 6 ijms-24-12021-f006:**
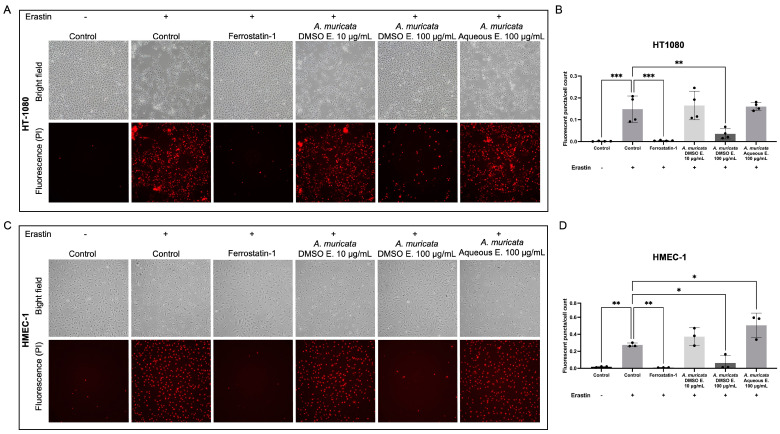
*Annona muricata* DMSO extract protects cells from ferroptosis. (**A**) Representative images of HT-1080 (**A**) and HMEC-1 (**C**) in the control condition and treated with the ferroptosis-inducer erastin, alone or in cotreatment with ferrostatin-1 (rescues from ferroptosis, used as a positive control), *A. muricata* DMSO extract (10 or 100 µg/mL), or *A. muricata* aqueous extract (100 µg/mL). Cells were also stained with propidium iodide to check membrane integrity, so bright field and fluorescence images are shown. Quantification of the fluorescent puncta, meaning cells with damaged membrane, normalized by cell count of HT-1080 (**B**) and HMEC-1 (**D**) cells. Means ± SD of a minimum of three independent experiments are shown, and one-way ANOVA + Dunnett’s multiple comparisons tests were performed for the statistical analysis. (* *p* < 0.05; ** *p* < 0.01; *** *p* < 0.001).

## Data Availability

Data are available within the article and Appendix A.

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
