# Peer review of "A Proteomic Study of the Bioactivity of Annona muricata Leaf Extracts in HT-1080 Fibrosarcoma Cells"

_ijms, 2023, doi:10.3390/ijms241512021_

Round 1

Reviewer 1 Report

The presented paper is focused on the study of mechanism of action of the aqueous and DMSO extracts from Annona muricata (Graviola) leaves. This American tropical plant is well-known for its numerous traditional ethnobotanic uses and pharmacologic applications. This makes the theme rather interesting.

The researchers performed proteomic analysis of HT-1080 human fibrosarcoma cells after the extracts' impact as well as of media where these cells were cultured. As a result, the spectrum of deregulated proteins was revealed.

This study should be considered as a pilot one because the presented data do not give us opportunity to make serious conclusions about the real response of the protein components to the biologic activity of the studied extracts. Also, it needs to be proved that changes in the gene spectrum play a significant role in this response.

It should also be noted that the performed research with the used HT-1080 and HMEC-1 cell lines is not quite adequate for reliable conclusions useful for further clinical study. More tumor and normal cultured cell lines are needed as objects of the investigation. This is the reason why we would support the authors in their conclusion about the necessity of further studying the physiologic activity of Annona muricata extracts and their mechanisms of action.

It needs to be mentioned here that the used methods are modern and adequate, references are consistent with the theme of the work and reflect the nowadays matter of affairs in the area of investigation, and the illustrative material, both in the main text and in supplements, helps to understand the obtained results.

Minor editing of English language is required because of some mistakes in syntax.

Author Response

Thank you for your kind comments. The conclusion has been modified to highlight that our research is preliminary, and thus more studies are needed.

It should also be noted that the performed research with the used HT-1080 and HMEC-1 cell lines is not quite adequate for reliable conclusions useful for further clinical study. More tumor and normal cultured cell lines are needed as objects of the. This is the reason why we would support the authors in their conclusion about the necessity of further studying the physiologic activity of Annona muricata extracts and their mechanisms of action. investigation

Thank you, this has also been included in the discussion.

It needs to be mentioned here that the used methods are modern and adequate, references are consistent with the theme of the work and reflect the nowadays matter of affairs in the area of investigation, and the illustrative material, both in the main text and in supplements, helps to understand the obtained results. Thank you for your constructive comments.

Reviewer 2 Report

The manuscript „A Proteomic Study of the Bioactivity of Annona Muricata Leaf  Extracts in HT-1080 Fibrosarcoma Cells” presents an interesting study on effects of graviola leaf extracts on selected cell lines.

The approach steps further than typical studies on natural products, as not only the direct activity is reported, but changes in proteome are investigated as well, indicating possible molecular targets of extract components, which are further investigated to confirm the discovered effects.

The discussion is well prepared and contains sufficient introductory information commenting the results to comfortably read and understand the material. However, this is a benefit obtained only after reading a Results part, which provides data and some conclusions in a really concise way. After finishing, I returned to the Introduction, but the aim of work was brief, probably explaining why the reading of the Results part was not comfortable, at least in my opinion. I understand that describing ferroptosis in Introduction is not feasible, as the process is considered as a result of the study. Still, some information of cell line selection and general plan of study may be beneficial before Results part.

The identification of extract components was described in another paper (7). As in this study a difference in activity was observed, do the Authors predict which components may be the source of activity? Are there any of the detected compounds known for similar effects?

There are some questions that affect particular parts of the text.

Table 1 presents limited amount of data, maybe the information could be combined with Figure 1?

The concentration representation in Figure 1 is worth mentioning as an interesting example of data presentation.

Some figure captions contain a lot of information. In case of figure 3, in my opinion , some information could be moved to regular text, as readers frequently omit figure captions. Explanation for STRING procedure is mentioned in caption.

Figure 6. Is the fluorescence very weak or the images look dark in my pdf copy? Without significant enlargement, the effects are practically invisible.

Line 181, line 225: please explain “movements”: Regarding the movements of cell cycle kinases CDK1 and 4

Line 266: this fragment suggests that the DMSO extract works with the mentioned antioxidant systems, please clarify and provide reference.

Part 4.5: the volumes are presented as “uL”: in 200 uL of 7 M urea. Please verify this unit and correct if needed. The same unit is used in 4.6

part 4.7: please verify the units (column diameter, particle size, pores): Acclaim PepMap 100, 75 um x 2cm, C18, 3 um, 100 A; PepMap RSLC C18, 2 um, 100 A, 75 um x 50 cm, (flow unit) flow rate of 20 uL/min

Consider using m/z in italics

Line 396: MS/MS2 – is it combination of MS and MS2? Please consider using one type of notation (MS/MS or MS2) in MS description

Not all reagents are described with source, some abbreviations in Materials are not explained.

Table S5 caption suggests that the compounds were detected in both extracts (identified by us in both aqueous and DMSO Annona muricata leaf extracts), whereas the lists are from separate experiments (extractions) – is “both” necessary?

Minor issues:

Chapter titles generate problems with italics and Latin name of plant (2.1)

Please check leaf/leaves in the text, as in line 17 there is a phrase: these leave extracts

Line 22: DMSO extracts protect from ferroptosis to both HT-1080 fibrosarcoma cells and HMEC-1

Line 30: plant is missing? Graviola (Annona muricata) is an American tropical that produces

Line 142: please verify Go/G1, G0/G1 and G0/G1 (line 435) in the text

line 261: please check: since it does not induce it

Line 303: please unify the centrifuge speed presentation (13000 in 303; 14,000 in 342)

Line 327: HCl-2-propanol – please check the final format for loss of second hyphen

line 331: please specify control conditions

part 4.6: Ammonium bicarbonate – is capital letter really needed?

Line 463: aquose extract (?)

some comments are included in the report

Author Response

The manuscript „A Proteomic Study of the Bioactivity of Annona Muricata Leaf  Extracts in HT-1080 Fibrosarcoma Cells” presents an interesting study on effects of graviola leaf extracts on selected cell lines.

Thank you for your comments.

The approach steps further than typical studies on natural products, as not only the direct activity is reported, but changes in proteome are investigated as well, indicating possible molecular targets of extract components, which are further investigated to confirm the discovered effects.

Thank you for your appreciation.

The discussion is well prepared and contains sufficient introductory information commenting the results to comfortably read and understand the material. However, this is a benefit obtained only after reading a Results part, which provides data and some conclusions in a really concise way.

Again, thank you for your appreciation.

After finishing, I returned to the Introduction, but the aim of work was brief, probably explaining why the reading of the Results part was not comfortable, at least in my opinion. I understand that describing ferroptosis in Introduction is not feasible, as the process is considered as a result of the study. Still, some information of cell line selection and general plan of study may be beneficial before Results part.

This comment is very reasonable and we think that you are right. The introduction has been modified, and now information about the selected cell lines and the study plan are more accurately explained. 

The identification of extract components was described in another paper (7). As in this study a difference in activity was observed, do the Authors predict which components may be the source of activity? Are there any of the detected compounds known for similar effects?

Thank you for this commentary, as it is a very good point. A discussion about the molecules that we (and other groups) detected in the Graviola extract, and the evidence of activity of some of these isolated compounds, has been included in the discussion. 

There are some questions that affect particular parts of the text.

Table 1 presents limited amount of data, maybe the information could be combined with Figure 1?

Table 1 has been eliminated, and this information has been included in Figure 1.

The concentration representation in Figure 1 is worth mentioning as an interesting example of data presentation.

Thank you.

Some figure captions contain a lot of information. In case of figure 3, in my opinion , some information could be moved to regular text, as readers frequently omit figure captions. Explanation for STRING procedure is mentioned in caption.

We agree that figure caption 3 is rather extense. However, we have tried to shorten it, and we have realized that removing information would result in some elements of the figure not being properly explained. 

Figure 6. Is the fluorescence very weak or the images look dark in my pdf copy? Without significant enlargement, the effects are practically invisible.

Thank you, the contrast and brightness of figure 6 has been modified for better visualization.  

Line 181, line 225: please explain “movements”: Regarding the movements of cell cycle kinases CDK1 and 4

“Movements” has been changed to “changes in the expression”.

Line 266: this fragment suggests that the DMSO extract works with the mentioned antioxidant systems, please clarify and provide reference.

This has been modified as well, thank you.

Part 4.5: the volumes are presented as “uL”: in 200 uL of 7 M urea. Please verify this unit and correct if needed. The same unit is used in 4.6

This was an error, it has been corrected.

part 4.7: please verify the units (column diameter, particle size, pores): Acclaim PepMap 100, 75 um x 2cm, C18, 3 um, 100 A; PepMap RSLC C18, 2 um, 100 A, 75 um x 50 cm, (flow unit) flow rate of 20 uL/min

This has been corrected.

Consider using m/z in italics

This has been corrected.

Line 396: MS/MS2 – is it combination of MS and MS2? Please consider using one type of notation (MS/MS or MS2) in MS description

This has been corrected, and now is consistently named MS/MS, thank you.

Not all reagents are described with source, some abbreviations in Materials are not explained.

This has been corrected.

Table S5 caption suggests that the compounds were detected in both extracts (identified by us in both aqueous and DMSO Annona muricata leaf extracts), whereas the lists are from separate experiments (extractions) – is “both” necessary?

This has been corrected.

Minor issues:

Chapter titles generate problems with italics and Latin name of plant (2.1)

Please check leaf/leaves in the text, as in line 17 there is a phrase: these leave extracts

Line 22: DMSO extracts protect from ferroptosis to both HT-1080 fibrosarcoma cells and HMEC-1

Line 30: plant is missing? Graviola (Annona muricata) is an American tropical that produces

Line 142: please verify Go/G1, G0/G1 and G0/G1 (line 435) in the text

line 261: please check: since it does not induce it

Line 303: please unify the centrifuge speed presentation (13000 in 303; 14,000 in 342)

Line 327: HCl-2-propanol – please check the final format for loss of second hyphen

line 331: please specify control conditions

part 4.6: Ammonium bicarbonate – is capital letter really needed?

Line 463: aquose extract (?)

All these minor issues have been corrected. Thank you very much for your time and thorough corrections. 

Reviewer 3 Report

The authors studied the effect of Graviola (Annona muricata) plant extract on protein expression in HT-1080 Fibrosarcoma Cells using the proteomic method. The authors studied the extracellular proteins from the HT-1080 Fibrosarcoma Cells. It is not clear why the authors quantified the extracellular proteins and not cellular protein expression. There is not enough background on the study and biological relevance to proteomics.

In the abstract the authors state that “we use a proteomic study to reveal new bioactivities of these leaf extracts on both conditioned media and extracts…” it is not clear how proteomic study helped in bioactivity assays or improved the bioactivity of the plant extract.

Figure 2 does not convey any information as it is poorly labeled. How many and which proteins were up and down expressed? What was the control? Same for Figure 3- how these proteins appeared here? 0.5 fold change is not significant, it is an equivalent expression. In fact, all figures are poorly labeled and hard to understand.

The authors mentioned in the methods “Both proteins from culture media and cell extracts were purified…” This is unclear if the authors are studying extracellular or cellular proteins.  Do these experiments were done in duplicates or triplicates? Does the data files were searched with the plant protein database to remove any background?

Label-free quantification and differential expression seem very preliminary and should have more details.

Extensive English editing is needed. 

Author Response

The authors studied the effect of Graviola (Annona muricata) plant extract on protein expression in HT-1080 Fibrosarcoma Cells using the proteomic method. The authors studied the extracellular proteins from the HT-1080 Fibrosarcoma Cells. It is not clear why the authors quantified the extracellular proteins and not cellular protein expression. There is not enough background on the study and biological relevance to proteomics.

In our work we quantify extracellular (in culture media) and intracellular (in cell extracts) proteins.

In the abstract the authors state that “we use a proteomic study to reveal new bioactivities of these leaf extracts on both conditioned media and extracts…” it is not clear how proteomic study helped in bioactivity assays or improved the bioactivity of the plant extract.

We thank Reviewer#3 for his/her comment. According to it, in the amended manuscript we have modified the abstract including the sentence: “In the present study, we use a proteomic approach to detect altered pattern in proteins on both conditioned media and extracts of HT-1080 fibrosarcoma under treatment conditions, revealing new potential bioactivities of Annona muricata extracts.”

Figure 2 does not convey any information as it is poorly labeled. How many and which proteins were up and down expressed?

As indicated in the figure caption, proteins were considered upregulated (the ones in the red box) or downregulated (the ones in the green box) if fold-change values were > 2 (circles labeled in red) or < 0.5 (circles labeled in green), respectively, with a p-value < 0.01. 
As indicated in the text, the lists of all significantly over- and under-expressed proteins are shown in Supplementary Material as Tables S1 to S4.

What was the control?

We apologize for not indicating the complete description of the analysis method in caption of Figure 2. Now, the amended manuscript includes the sentence: “Volcano plots show deregulated proteins in HT-1080 cell extracts and conditioned media after treatment with A. muricata aqueous and DMSO extracts, compared to control (non-treated cells).” According to this, each circle in the volcano plots represents a ratio: the mean abundance of a protein in the treated cells relative to the mean abundance of that same protein in the untreated control.

Same for Figure 3- how these proteins appeared here?

This is detailed extensively in the figure caption.

0.5 fold change is not significant, it is an equivalent expression. In fact, all figures are poorly labeled and hard to understand.

0.5 is not a p-value, it is the fold change. We have considered significant all those differentially expressed proteins whose p-value was below 0.01, as indicated in the figure captions, and as it can be deduced by looking at the Y axis of the graph: Log10 p-value=2; then: p-value=0.01.

The authors mentioned in the methods “Both proteins from culture media and cell extracts were purified…” This is unclear if the authors are studying extracellular or cellular proteins.  

We apologize for the mistake in the redaction. In the amended manuscript this is corrected: “Proteins from both culture media and cell extracts were independently purified by…”
As mentioned in the abstract, in the present study, we carry out a proteomic study on both conditioned media and extracts of HT-1080 fibrosarcoma cells. We did study both, extracellular and intracellular proteins.

Do these experiments were done in duplicates or triplicates?

Proteomic experiments were done in quadruplicate. 

Does the data files were searched with the plant protein database to remove any background?

To remove any background, two databases of plant contaminants were used: the UniProt  Annoma muricata database (117 sequences, version 2023-05-03) and the UniProt Arabidopsis Thaliana database (119635 sequences, version 2021-09-01). No proteins were detected in the A. muricata database, and none of the proteins detected in the A. thaliana database were common to those detected in the human database. 

Label-free quantification and differential expression seem very preliminary and should have more details.

We have tried to offer a rigorous contribution to the study of Annona muricata extracts on HT-1080 fibrosarcoma cell secreted and endogenous proteins. Of course, much more remains to be discovered but we are confident that our results add interesting and novel knowledge.

We have carried out an extensive editing of English, as requested.

Thank yo for your comments and suggestions.

Round 2

Reviewer 3 Report

The revised version of the manuscript looks much improved.

As per my earlier comment, “Figure 2 does not convey any information as it is poorly labeled.” Mention the Log2 ratio of which samples are on the x-axis. Label the widely located protein spots as it is labeled in 2B. “How many and which proteins were up and down expressed?” Write on both sides how many proteins are up and downregulated.

Figure 3 STRING network stats is not useful and could be moved the supporting information. Plot the Biological processes with their FDR level to quickly assess. Figure 3 caption is highly extended instead cite their original reference.

Minor English editing is needed.

Author Response

Thank you for your evaluation of the improvements made in our manuscript.

Figures 2 and 3 have been changed according to your suggestions. We now include a Table in figure 3 with only the FDR levels of the biological processes (which has made us possible to reduce Figure 3 caption extension) and we now present the complete network stats as an additional Supplementary Figure S1.

We have also changed Figure 4 in response to a suggestion made by the editors.

We have revised and edited English, as suggested.